# Soluble Endoglin Stimulates Inflammatory and Angiogenic Responses in Microglia That Are Associated with Endothelial Dysfunction

**DOI:** 10.3390/ijms23031225

**Published:** 2022-01-22

**Authors:** Eun S. Park, Sehee Kim, Derek C. Yao, Jude P. J. Savarraj, Huimahn Alex Choi, Peng Roc Chen, Eunhee Kim

**Affiliations:** Vivian L. Smith Department of Neurosurgery, McGovern Medical School, The University of Texas Health Science Center at Houston, 6431 Fannin Street, Houston, TX 77030, USA; Sehee.Kim@uth.tmc.edu (S.K.); Derek.C.Yao@uth.tmc.edu (D.C.Y.); Jude.P.Savarraj@uth.tmc.edu (J.P.J.S.); Huimahn.A.Choi@uth.tmc.edu (H.A.C.); Peng.R.Chen@uth.tmc.edu (P.R.C.)

**Keywords:** brain arteriovenous malformation (bAVM), soluble endoglin (sENG), microglia, endothelial cells (ECs), inflammation, angiogenesis

## Abstract

Increased soluble endoglin (sENG) has been observed in human brain arteriovenous malformations (bAVMs). In addition, the overexpression of sENG in concurrence with vascular endothelial growth factor (VEGF)-A has been shown to induce dysplastic vessel formation in mouse brains. However, the underlying mechanism of sENG-induced vascular malformations is not clear. The evidence suggests the role of sENG as a pro-inflammatory modulator, and increased microglial accumulation and inflammation have been observed in bAVMs. Therefore, we hypothesized that microglia mediate sENG-induced inflammation and endothelial cell (EC) dysfunction in bAVMs. In this study, we confirmed that the presence of sENG along with VEGF-A overexpression induced dysplastic vessel formation. Remarkably, we observed increased microglial activation around dysplastic vessels with the expression of NLRP3, an inflammasome marker. We found that sENG increased the gene expression of VEGF-A, pro-inflammatory cytokines/inflammasome mediators (TNF-α, IL-6, NLRP3, ASC, Caspase-1, and IL-1β), and proteolytic enzyme (MMP-9) in BV2 microglia. The conditioned media from sENG-treated BV2 (BV2-sENG-CM) significantly increased levels of angiogenic factors (Notch-1 and TGFβ) and pERK1/2 in ECs but it decreased the level of IL-17RD, an anti-angiogenic mediator. Finally, the BV2-sENG-CM significantly increased EC migration and tube formation. Together, our study demonstrates that sENG provokes microglia to express angiogenic/inflammatory molecules which may be involved in EC dysfunction. Our study corroborates the contribution of microglia to the pathology of sENG-associated vascular malformations.

## 1. Introduction

Brain arteriovenous malformations (bAVMs) are an abnormal tangle of enlarged vasculature in the brain caused by a direct connection of arteries and veins without an intervening capillary bed [1,2]. Although bAVMs are rare (about one in 2000–5000 people), they are the major cause of intracerebral hemorrhages in children and young adults. Brain AVMs can also cause other neurological symptoms such as seizures, headaches, and difficulty in movement, speech, and vision [1,3,4,5]. Currently, the pathophysiology of bAVMs is poorly understood, and the treatment options for bAVMs are limited, mostly relying on surgical resection, endovascular embolization, or radiation.

Endoglin (ENG) is a type I transmembrane glycoprotein which is mainly expressed in vascular endothelial cells (ECs) and activated monocytes [6]. ENG modulates transforming growth factor-β (TGFβ) superfamily signaling and regulates endothelial quiescence and angiogenesis, and its mutation results in hereditary hemorrhagic telangiectasia type 1 (HHT1), an autosomal dominant vascular disorder [7,8]. Previous studies have shown that soluble ENG (sENG), which is a circulating cleaved form of ENG, was related to the development of sporadic bAVMs [9]. For instance, elevated plasma sENG levels were observed in human bAVM patients [9]. A focal overexpression of sENG with overexpression of the vascular endothelial growth factor (VEGF)-A has been observed to induce vessel dysplasia in mice [9]. However, the exact mechanism of sENG-induced abnormal vascular formation remains to be clarified. 

Increased inflammation can be observed in bAVMs [10,11,12,13]. Enhanced accumulation of microglia has been observed around bAVMs in both human patients and mouse models of bAVMs [12,13]. In addition, increased inflammatory cytokines have been observed in bAVM tissues as well as in the patient’s blood [12,13]. This evidence suggests that inflammation is critical in bAVM pathophysiology. Meanwhile, pro-inflammatory activity of sENG has been shown in mouse brain. Soluble ENG enhanced oxidative stress and activity of matrix metalloproteinases (MMP-2 and MMP-9), the mediators of inflammatory response, in sENG/VEGF-A-overexpressing mouse brain [9]. Studies have also shown that sENG increases nuclear factor-kappa B (NFkB) activation and interleukin-6 (IL-6) expression in cultured ECs [14]. 

Based on the roles of sENG and microglia in inflammation and bAVMs, we hypothesized that sENG-stimulated microglia are involved in vascular abnormality by means of inflammatory and angiogenic responses. In this study, we found that systemic sENG administration induced dysplastic capillaries over the VEGF-A overexpression. We also observed an increase in activated microglia expressing NLRP3, an inflammasome marker, around the abnormal vessels. In vitro studies further revealed that sENG-stimulated BV2 microglia expressed inflammatory and angiogenic factors, and the conditioned medium from sENG-treated BV2 significantly increased angiogenic markers and migration/tube formation in ECs. These results suggest that microglia may contribute to sENG-induced endothelial dysfunction via expressing inflammatory and angiogenic factors. The microglia-mediated endothelial dysfunction can be a possible mechanism underlying sENG-induced bAVMs.

## 2. Results

### 2.1. Soluble ENG Induced Vascular Dysplasia in the Mouse Brain with VEGF-A Overexpression 

A previous study showed that focal brain overexpression of VEGF-A and sENG resulted in the formation of dysplastic vessels [9]. Since enhanced circulating sENG has been detected in bAVM patients, we systemically injected recombinant human sENG into mice with focal brain VEGF-A overexpression by intracerebral injection of AAV1-VEGF-A (Figure 1A). Blue latex-cast brains clearly showed dysplastic and enlarged vasculatures in mice with sENG treatment and VEGF-A overexpression. The dysplastic vessels formed around the AAV1-VEGF-A-injected site (ipsilateral) (Figure 1B), and the volume of dysplastic vessels was significantly increased compared to normal vessels in the other hemisphere without AAV1-VEGF-A injection (contralateral) (Figure 1B,C). Dysplastic vessels were not found in mouse brains with VEGF-A overexpression alone, that is, without sENG injection or sENG treatment with a control AAV1-LacZ injection (Figure 1C). By immunostaining using the CD31 antibody, we confirmed the presence of significantly enlarged vessels in the brains of mice with VEGF-A overexpression and sENG administration (Figure 1D). These data suggest that the combination of sENG with VEGF-A overexpression causes vascular malformations.

### 2.2. Microglia with Enhanced Inflammasome Marker Were Recruited around sENG/VEGF-A-Induced Dysplastic Vessels in Mice

Brain AVM patients and mouse models displayed microglial activation and increased inflammatory markers in bAVM tissues [11,12,15,16]. We determined the microglial activation around sENG/VEGF-A-induced dysplastic vessels by means of Iba-1 immunostaining. Iba-1-positive (+) activated microglia were localized around AAV1-VEGF-A-injected lesions (Figure 2A) with a significantly higher fluorescence intensity of Iba-1, indicating a thick cell body compared to the one in control mice that were administrated AAV1-VEGF-A or sENG alone. Conversely, the microglia in control mice had small cell bodies and long branches (Figure 2B) [17,18,19]. 

In preeclampsia patients who have a high level of sENG and inflammatory cytokines in their plasma, NLRP3-mediated inflammasome activation has been shown in their placenta [20]. To test if sENG/VEGF-A activates inflammasome in the microglia around dysplastic vessels, we performed immunostaining using NLRP3 with Iba-1 and CD31 antibodies. The accumulated microglia around dysplastic vessels showed remarkable NLRP3+ signals compared to the normal vessels (Figure 2A). In addition, the fluorescence intensity of NLRP3 was significantly increased in the activated microglia around the dysplastic vessels (Figure 2A,C). The data suggest the potential role of activated microglia with inflammasome activation in sENG/VEGF-A-induced vascular malformation. 

### 2.3. Soluble ENG Induced Expression of VEGF-A and Inflammatory/Inflammasome Markers in Cultured Microglia

To test if sENG stimulates microglia to enhance inflammatory/inflammasome activation, we measured the inflammatory/inflammasome markers in cultured BV2 microglia treated with sENG. The results showed that sENG significantly increased the mRNA levels of VEGF-A, tumor necrosis factor (TNF)-α, IL-6, and MMP-9 (Figure 3A), and inflammasome markers, including NLRP3, apoptosis-associated speck-like protein containing a caspase recruitment domain (ASC), caspase-1, and IL-1β (Figure 3B). The data demonstrate that sENG induces inflammatory/inflammasome responses in microglia. 

### 2.4. Soluble ENG-Stimulated Microglia Modulated Angiogenic Factors in Endothelial Cells

Uncontrolled excessive angiogenesis is considered a hallmark of bAVM development [21], and it has been shown that microglia-derived angiogenic and inflammatory mediators are associated with enhanced angiogenesis in ECs [22]. To test if sENG-treated microglia regulate the angiogenic properties of ECs, we measured the expression of TGFβ, Notch-1, and phospho (p)-ERK1/2 – all of which are angiogenic factors that have been related to bAVM development [16,23,24,25,26,27,28,29]—in ECs treated with BV2-conditioned media (BV2-sENG-CM). BV2-sENG-CM significantly increased TGFβ and Notch-1 gene expression in ECs compared to controls treated with vehicle or sENG only (Figure 4A). In addition, BV2-CM (the CM from BV2 without sENG treatment) significantly increased p-ERK1/2 levels in ECs, and BV2-sENG-CM further enhanced the p-ERK1/2 levels. The direct treatment of sENG to ECs also induced ERK1/2 activation; however, the levels were lower than those in ECs treated with BV2-sENG-CM (Figure 4B). We further determined the expression of IL-17 receptor D (IL-17RD), which is downregulated in a high angiogenic niche such as cancer [30,31]. BV2-sENG-CM significantly decreased the level of IL-17RD in ECs while direct treatment of sENG did not affect the levels in ECs (Figure 4C). These results suggest that microglial expressing molecules mediate sENG-induced angiogenic response in ECs.

### 2.5. Soluble ENG-Stimulated Microglia Modulated Endothelial Cell Functions

Altered EC functions such as enhanced proliferation, migration, or tube formation have been associated with pathologic angiogenesis and abnormal vessel formation [32,33,34]. We therefore tested whether sENG-treated BV2 regulates EC functions. In a scratch-wound assay, BV2-sENG-CM treatment showed faster EC migration compared to sENG- or BV2-CM-treated ECs (Figure 5A). By means of a tube formation assay using Matrigel, we also confirmed that BV2-sENG-CM significantly increased the total tube length and the branch and loop counts in ECs compared to the controls, suggesting enhanced tube formation (Figure 5B). The data demonstrate that sENG-treated microglia modulate EC functions. 

## 3. Discussion

The present study demonstrates that microglia were activated around dysplastic/enlarged vessels induced by systemic sENG injection with focal overexpression of VEGF-A. We also found that sENG increased angiogenic and inflammatory/inflammasome molecules in cultured microglia and, ultimately, showed that sENG-treated microglia regulated the angiogenic properties of ECs. This is the first study to demonstrate the association of microglia in sENG-induced endothelial dysfunction, providing a novel insight into the inflammatory response in bAVM pathology.

The TGFβ signaling pathway is essential for vessel development and maintenance [35,36]. The impairment of the TGFβ signaling pathway is associated with cerebrovascular disease, including AVMs and aneurysms [25]. In particular, the haploinsufficiency of TGFβ receptors such as Eng- or Alk1 with a focal VEGF-A overexpression caused the development of HHT type 1 or 2-associated AVMs in mice [37,38]. Meanwhile, enhanced activity of MMPs (e.g., MMP-14) has been observed in bAVMs and may be involved in the sENG production via ENG cleavage [9,39,40,41]. Although the exact mechanism by which sENG disrupts TGFβ signaling remains to be clarified, it is known that sENG can bind to circulating TGFβ or BMP9 (the ligands of Alk1) and function as a decoy receptor causing reduced activation of TGFβ pathway (Figure 6A) [42,43]. Supportively, there is some, albeit limited, evidence that sENG:BMP9 and sENG:TGFβ complex occur in the plasma of preeclampsia patients [43,44,45].

Elevated sENG levels have been seen in human bAVM tissues, and the focal overexpression of sENG and VEGF-A by a virus system has been observed to induce vascular dysplasia [9]. Based on these findings, our study tested whether the systemic administration of sENG leads to abnormal vascular development. We confirmed that recombinant sENG consistently induced vascular dysplasia in the focal area of mouse brain injected with AAV-VEGF-A (Figure 1). Our results showed that neither VEGF-A overexpression without sENG nor sENG administration without VEGF overexpression induced dysplastic vessel development (Figure 1B,C) suggesting that both defective ENG function and VEGF-A overexpression are required in abnormal vascular development [46].

The mechanism in which sENG/VEGF-A induces vascular dysplasia is unclear. While high levels of sENG have been identified in several human pathological conditions related to dysregulated vascular permeability, endothelial function, and/or angiogenesis (preeclampsia, coronary atherosclerosis, Alzheimer’s disease, hypertension and diabetes) [43,47,48,49], our study showed microglial activation with enhanced NLRP3, an inflammasome marker, around sENG/VEGF-induced dysplastic vessels (Figure 2). This suggests that microglia play a role in abnormal vascular development. Supportively, our in vitro study confirmed that sENG increased VEGF-A, inflammatory cytokines, and inflammasome markers in BV2 microglial cells (Figure 3). We also found that BV2-sENG-CM treatment enhanced the expression of angiogenic mediators (Figure 4) and functions in ECs (Figure 5). Taken together, our in vivo and in vitro data suggest that microglia upon increased circulating sENG may accelerate the abnormal function of ECs by expressing inflammatory and angiogenic factors and contribute to abnormal vascular development. Further studies should be undertaken to define the detailed molecular mechanism underpinning the role of microglia in mediating sENG-induced EC dysfunction and vascular malformation by identifying and targeting the molecular factor(s) released from sENG-stimulated microglia.

The effect of sENG alone on cultured BV2 is apparent in causing NLRP3 expression (Figure 3B). However, no NLRP3-positive cell was detected upon sENG treatment in mouse brain (Figure 2A). Although we did not measure sENG levels in the mice after the injection of recombinant sENG (4 μg/kg, around 0.12 μg/30 g of mouse), the circulating sENG would be less than 0.067 μg/mL if total mouse blood is calculated as 1.8 mL. Therefore, we assume that the amount of sENG used to treat the BV2 cells (0.5 or 1 μg/mL) was much higher than the physiologic sENG levels in our mice after the systemic injection of sENG. In addition, the effect of sENG on microglia in mouse brain could be affected by other factors, such as the vascular constituents and other cells. However there was a direct impact of sENG on microglia in in vitro. Based on that, it may be that the effect of sENG alone on microglia in mouse brain is weaker or slower than it would be on cultured microglia.

Our findings suggest that the sENG stimulation of microglia has a pro-angiogenic effect on ECs. However, a previous study has shown that sENG alone can inhibit endothelial tubulogenesis and cell migration [50]. While the study showed that 100 ng/mL of sENG reduced EC migration in the scratch wound assay [50], our study demonstrated that 1000 ng/mL of sENG did not reduce EC migration. In addition, BV2-sENG-CM exhibited an even faster migration and higher tube formation in ECs (Figure 5). Therefore, it remains to be determined whether EC functions are differentially regulated depending on sENG concentration. Nevertheless, our results clearly show that microglia mediate the effect of sENG in EC functions. With previous evidence demonstrating the role of microglia in vascular architecture via regulating TGFβ signaling in ECs [51], our study suggests that sENG-induced microglial activation may be sufficient to compensate for the sENG-induced loss of TGFβ signaling in ECs. This is partially supported by the increased TGFβ expression in ECs treated with BV2-sENG-CM (Figure 6B).

Microglia are involved in a broad range of brain functions, including control of neuronal synapse development, normal myelinogenesis or oligodendrocyte progenitor maintenance, phagocytosis of apoptotic cells, and brain repair [52,53,54,55,56,57]. However, their role in bAVM pathology is not yet clear. The capillary-associated microglia interact closely with the ECs, astrocytes, and pericytes that comprise the brain’s microvascular unit and blood-brain barrier (BBB) [58,59,60]. In a pathologic condition such as bAVM, damaged ECs can be the source of sENG [9]. The sENG shed from damaged ECs may stimulate microglia, and the factors from activated microglia (including angiogenic/inflammatory mediators) would subsequently lead to EC dysfunction. The expression of ENG by microglia/macrophages has been reported in healthy subjects and those with neurodegenerative diseases including Alzheimer’s disease and Parkinson’s disease [61,62,63,64]. Therefore, sENG can also be generated from microglia (besides ECs), and it is possible for sENG to have autocrine effects on microglia. However, it is not clear how sENG stimulates microglia. The mechanisms by which sENG activates the microglia remain to be investigated.

Targeting immune cells is an emerging approach in tackling cerebrovascular diseases [65]. Even though obvious inflammatory reactions and immune cell accumulation have been observed in human and mouse bAVMs [11,12,16], the significance of immune cells in bAVM pathophysiology has been overlooked. Our study provides evidence that microglia contribute to sENG-induced EC dysfunction and supports the idea that microglia play a critical role in bAVM pathology. Further studies elucidating the detailed mechanisms underlying the role of microglia in EC dysfunction will ultimately provide novel insights useful in the development of therapeutic strategies for bAVM patients by targeting microglia and the associated inflammation.

## 4. Materials and Methods

### 4.1. Animals

All experiments were performed in accordance with the approved animal protocols of the Center for Laboratory Animal Medicine and Care (CLAMC), University of Texas Health Science Center, Houston. Male C57BL/6 mice (8 weeks of age, 23–25 g) were housed under a 12:12 h light: dark cycle at an ambient temperature of 22 °C. Water and mouse chow were available ad libitum. Young male mice were selected in this study because bAVM is more frequently diagnosed in young males [66,67].

### 4.2. Treatment with AAV1-VEGF-A and sENG in Mice

Adeno-associated viral (AAV)1-mediated vascular endothelial growth factor (AAV1-CMV-hVEGF-A, 2 μL, 2 × 10^9^ gc/mL, Vector Biolabs, Malvern, PA) or AAV1-LacZ (2 μL, 2 × 10^9^ gc/mL, Vector Biolabs) was stereotaxically injected into the right striatum (anteroposterior: 0.5 mm, mediolateral: 2.0 mm, and dorsoventral: 3.0 mm from bregma) according to the mouse brain atlas. At six weeks after the AAV1-VEGF-A injection, recombinant human sENG protein (4 μg/kg, 1097-EN-025, R&D Systems, Minneapolis, MN, USA) was injected subcutaneously daily for 14 days. PBS was injected as a control.

### 4.3. Systemic Latex Vascular Casting and Analysis

To visualize the brain vasculature, systemic latex dye perfusion was performed. Blue latex dye (BR80B, Connecticut Valley Biological, Southampton, MA, USA) was injected into the left ventricle of the heart. Brains were collected and fixed in 10% neutral buffered formalin solution (HT501320, Sigma-Aldrich, St. Louis, MO, USA). The brains were sequentially dehydrated with a series of methanol (A412-4, Fisher Scientific, Hampton, NH, USA) solutions (50%, 75%, 95% and 100% in each 24 h), and cleared in benzyl benzoate (AC105862500, ACROS organic, Fair Lawn, NJ, USA) and benzyl alcohol (AC148390010, ACROS organic), (1:1 ratio). Each single dysplastic lesion image was captured under 30× magnification using a light microscope (Zeiss SteREO Discovery. V12, Jena, Germany). To measure the dysplastic vessel size, the regions of interest (ROIs) were drawn with a spline contour and automatically quantified by a blinded expert using Zen 2.3 software (Zeiss).

### 4.4. Immunofluorescence Staining and Analysis

Animals were transcardially perfused with PBS and fixed with 10% formalin. The collected brains were sectioned (30-μm thick) for immunohistochemical staining. The sections were incubated overnight with the following primary antibodies: mouse anti-CD31 (1:1000, AF3628, R&D Systems, Minneapolis, MN, USA) for endothelial cells (ECs), rabbit anti-Iba-1 (1:1000, 019-19741, Wako, Richmond, VA, USA) for microglia, and goat anti-NLRP3 (1:200, MAB7578, R&D Systems) for inflammasome markers. After incubation with primary antibodies, tissues were incubated with 488-conjugated donkey anti-goat IgG (1:500, 705-545-147, Jackson ImmunoResearch, West Grove, PA, USA) for CD31, 647-conjugated donkey anti-rabbit IgG (1:500, 711-605-152, Jackson ImmunoResearch) for Iba-1, or Cy3-congugated donkey anti-rat IgG (712-165-150, Jackson ImmunoResearch) for NLRP3. Stained tissues were observed using fluorescence microscopy (Leica DM4000 B LED, LAS V4.12, Wetzlar, Germany). To quantify the intensity of CD31 and Iba-1, between two and four contralateral or ipsilateral images were acquired for each mouse under 40× fluorescence microscopy and the intensity was measured using LAS V4.12 software (Leica).

### 4.5. Cell Culture and sENG Treatment

BV2 murine microglia (CRL-2468, ATCC, Manassas, VA, USA) and BALB/C mouse primary brain microvascular ECs (BALB-5023, Cell biologics, Chicago, IL, USA) were cultured with a medium containing high glucose (CM002-050, GenDEPOT, Katy, TX, USA) 10% fetal bovine serum (F0901-050, GenDEPOT) and penicillin-streptomycin (CA005, GenDEPOT) at 37 °C in a 5% CO_2_ humidified incubator. The medium for BALB/C mouse primary brain microvascular ECs was added EC growth supplement (E2759, Sigma-Aldrich). The cells were seeded in 6-well plates (5 × 10^4^ cells/well)and treated with recombinant mouse sENG protein (1320-EN-025, R&D Systems) for 24 h. The sENG-treated BV2-conditioned media (sENG-BV2-CM) were mixed with EC medium (50% v/v) and incubated on the ECs.

### 4.6. EC Function Assays

For the scratch-wound assay, mouse primary brain microvascular ECs were seeded on a 24-well plate (2 × 10^5^ cells/well) in the absence or presence of sENG-BV2-CM and incubated at 37 °C in a 5% CO_2_ humidified incubator. After 10 h, images were captured under a microscope and the wound closure rate was analyzed using Ibidi software (Fitchburg, WI, USA).

To assess tube formation, mouse primary brain microvascular ECs were seeded on Matrigel (356234, Corning, NY, USA) in a 48-well plate (1 × 10^5^ cells/well). The ECs were incubated in the medium with or without sENG-BV2-CM at 37 °C in a 5% CO_2_ humidified incubator. After 8 h, the images were captured under a microscope and the tube formation analyzed using Ibidi software.

### 4.7. Quantitative Real-Time PCR

Total mRNA was harvested from each BV2 or EC using RNeasy mini kits (74104, Qiagen, Hilden, Germany) according to the manufacturer’s protocols, and used for cDNA synthesis using a commercial kit (High-Capacity cDNA Reverse Transcription Kits, 4368814, Applied Biosystems, Foster City, CA, USA). The cDNA was tested by means of a real-time PCR reaction (Applied Biosystems Quant 3 Studio qPCR) using a Fast SYBR^TM^ Green Master Mix (Applied Biosystems, 4385612). PCR primers were purchased from Sigma-Aldrich. The following primers were used:
VEGF-A(F)CTCACCAAAGCCAGCACATA
(R)AATGCTTTCTCCGCTCTGAATGFβ(F)GCCCTTCCTGCTCCTCATG
(R)CCGCACACAGCAGTTCTTCTCNotch-1(F)CTGAGGCAAGGATTGGAGTC
(R)GAATGGAGGTAGGTGCGAAGIL-6(F)TGGTACTCCAGAAGACCAGAGG
(R)AACGATGATGCACTTGCAGATNF-α(F)ATGGCCTCCCTCTCATCAGT
(R)TTTGCTACGACGTGGGCTACMMP-9(F)GCCGACTTTTGTGGTCTTCC
(R)TACAAGTATGCCTCTGCCAGCNLRP3(F)CTTCTAGCTTCTGCCGTGGTCTCT
(R)CGAAGCAGCATTGATGGGACACaspase-1(F)GTACACGTCTTGCCCTCATTATCTG
(R)TTTCACCTCTTTCACCATCTCCAGASC(F)CTGAGCAGCTGCAAACGACTAAA
(R)CTTCTGTGACCCTGGCAATGAGTIL-1β(F)CAACCAACAAGTGATATTCTCCATG
(R)GATCCACACTCTCCAGCTGCAGAPDH(F)GGAGTCAACGGATTTGGTCG
(R)GGAATCATATTGGAACATGTAAACC

### 4.8. Western Blotting

BV2 and ECs were harvested and lysed using a protein lysis buffer (R4100, GenDEPOT, 50 mM Tris–HCl, pH 7.4, 150 mM NaCl, 1% triton X-100, 2 mM EDTA, 0.5% Sodium deoxycholate, 0.1% SDS) containing a phosphatase inhibitor mixture (P3200-005, GenDEPOT) and a protease inhibitor mixture (P3100-005, GenDEPOT). The proteins were separated by means of electrophoresis in 4–20% ExpressPlus™ Page Gel (M42015, GenScript, Piscataway, NJ, USA) using Mini-PROTEAN Tetra Cell (1658004, Bio-Rad Laboratories, Hercules, CA, USA) and transferred to polyvinylidene difluoride membranes (1704273, Bio-Rad Laboratories) using a Trans-Blot Turbo Transfer System (1704150, Bio-Rad Laboratories). The following primary and secondary antibodies were used for the western blot in this study: p-ERK1/2 (Thr202/Tyr204) (1:1000, 9101, Cell Signaling Technology, Danvers, MA, USA), t-ERK (1:1000, 9102, Cell Signaling Technology), IL-17RD (1:1000, AF2276, R&D Systems) or β-actin (1:1000, sc-47778, Santa Cruz Biotechnology, Dallas, TX, USA). After incubating the blots for 2 h at 4 °C, secondary antibodies were used: rabbit IgG horseradish peroxidase-conjugated antibody (1:2000, HAF008, R&D Systems) or mouse IgG horseradish peroxidase-conjugated antibody (1:2000, HAF007, R&D Systems).

### 4.9. Statistical Analysis

All values are expressed as mean ± standard error (SEM). Statistical significance (*p* < 0.05 for all analyses) was assessed by means of Student’s *t*-test for comparisons between two groups, or one-way ANOVA followed by Tukey’s post-hoc comparison for multi-way comparisons. Statistical analyses were performed using GraphPad Software (San Diego, CA, USA).

## 5. Conclusions

Our study demonstrated that sENG/VEGF-A induces the formation of dysplastic vessels, accompanied by microglial activation with high expression of inflammasome markers in mouse brains. Our in vitro study confirmed that sENG induced the expression of angiogenic and inflammatory molecules in microglia and the CM from sENG-stimulated microglia enhanced the expression of angiogenic factors and functions in ECs. This is the first study to demonstrate that sENG-stimulated microglia induce hyper-angiogenic phenotypes of ECs, suggesting the role of microglia in mediating sENG-induced EC dysfunction and vascular malformation.

## Figures and Tables

**Figure 1 ijms-23-01225-f001:**
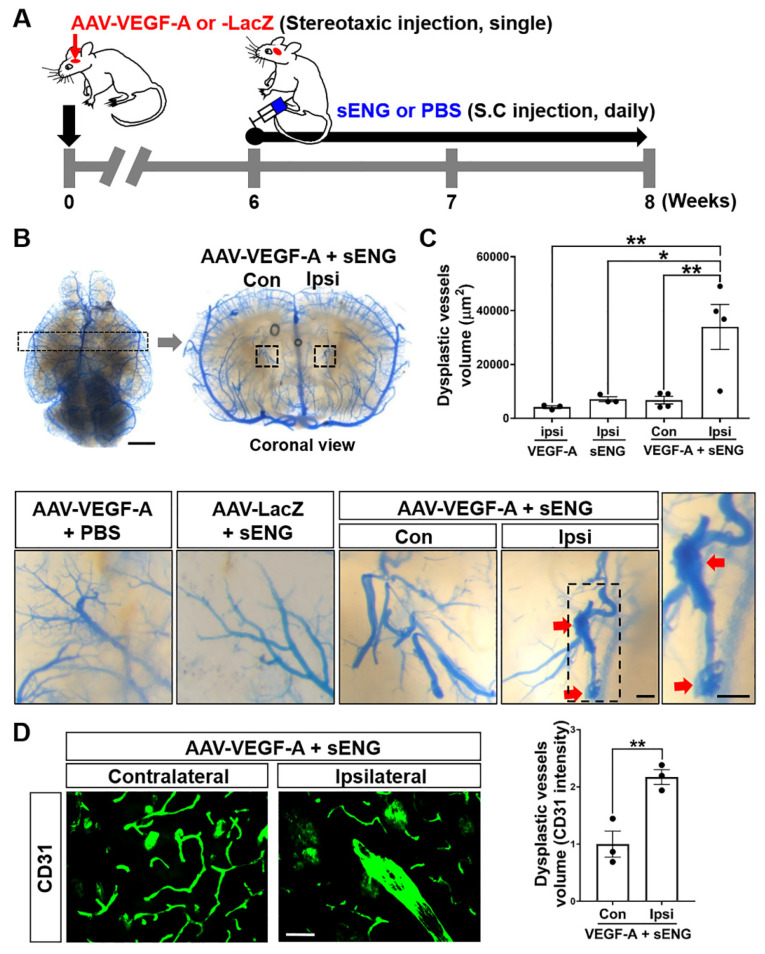
Soluble ENG/VEGF-A induces the formation of dysplastic vessels in the mouse brain. (**A**) Experimental scheme for sENG and AAV1-VEGF-A injection. The mice were stereotaxically injected with AAV1-VEGF-A (or AAV1-LacZ) into the intra-striatum and administered recombinant sENG (or PBS) subcutaneously (s.c.) every day for two weeks beginning at six weeks after the AAV1-VEGF-A injection. At eight weeks after the AAV1-VEGF-A injection, the mice were sacrificed. (**B**) Representative images of latex cast-clarified brains in mice injected with AAV1-VEGF-A + PBS, AAV1-LacZ + sENG, and AAV1-VEGF-A + sENG. Coronal sections showed the enlarged abnormal vasculature (inset, arrows) in the AAV1-VEGF-A injected site (ipsilateral) compared to the non-injected site (contralateral) in mice injected with AAV1-VEGF-A and sENG. Scale bars: 2 mm (whole brain image) and 100 μm (inset). (**C**) The quantification of vessel volume in mouse brains. AAV-VEGF-A (or VEGF-A), mice injected with vehicle (PBS) and AAV1-VEGF-A, sENG, mice injected with sENG and AAV1-LacZ, AAV-VEGF-A (or VEGF-A) + sENG, mice injected with AAV1-VEGF-A and sENG. *n* = 3–4, * *p* < 0.05, ** *p* < 0.01, One-way ANOVA. (**D**) Image of CD31 immunostained brain from mice injected with sENG and AAV1-VEGF-A and the quantification of CD31 intensity. Scale bar: 50 µm, *n* = 3, data are presented as mean ± SEM, ** *p* < 0.01, Student’s *t*-test. Con, contralateral, Ipsi, ipsilateral.

**Figure 2 ijms-23-01225-f002:**
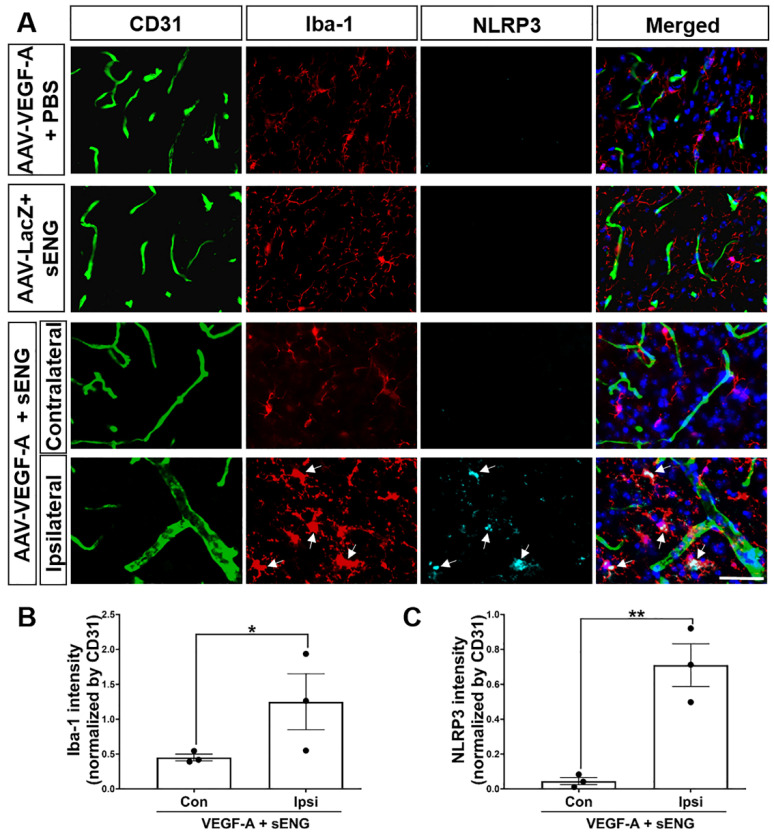
Soluble ENG/VEGF-A induced microglial activation with expression of inflammasome marker around dysplastic capillaries in mouse brain. (**A**) Immunostaining of mouse brain with CD31, Iba-1, and NLRP3 antibodies. Iba-1+ activated microglia are distributed around sENG/VEGF-induced dysplastic vessels in the mouse brain. Iba1+/NLRP3+ cells (arrows) indicate the inflammasome activation within the activated microglia. Scale bar: 50 μm. (**B**,**C**) The quantification of intensity of Iba-1 (**B**) and NLRP3 (**C**) fluorescence normalized by CD31, respectively. *n* = 3; data are presented as mean ± SEM; * *p* < 0.05, ** *p* < 0.01 vs. contralateral; Student’s *t*-test.

**Figure 3 ijms-23-01225-f003:**
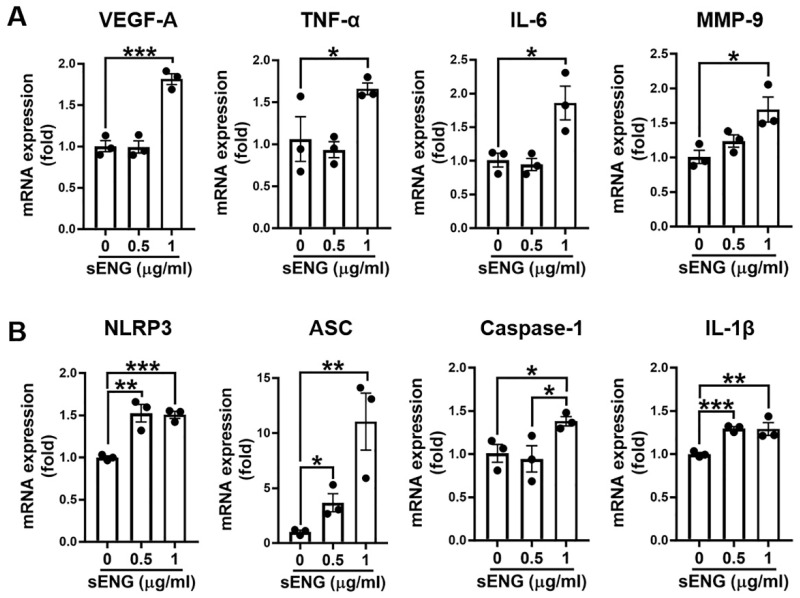
Soluble ENG induced gene expression of angiogenic and inflammatory mediators in microglia. (**A**,**B**) Change of gene expression levels of angiogenic mediator and inflammatory cytokines (**A**) and inflammasome markers (**B**) in BV2 microglia by sENG treatment. BV2 cells were stimulated with 0.5 or 1 μg/mL of sENG for 24 h. Each mRNA level was normalized with GAPDH. *n* = 3; data are presented as mean ± SEM; * *p* < 0.05, ** *p* < 0.01, *** *p* < 0.001; Student’s *t*-test.

**Figure 4 ijms-23-01225-f004:**
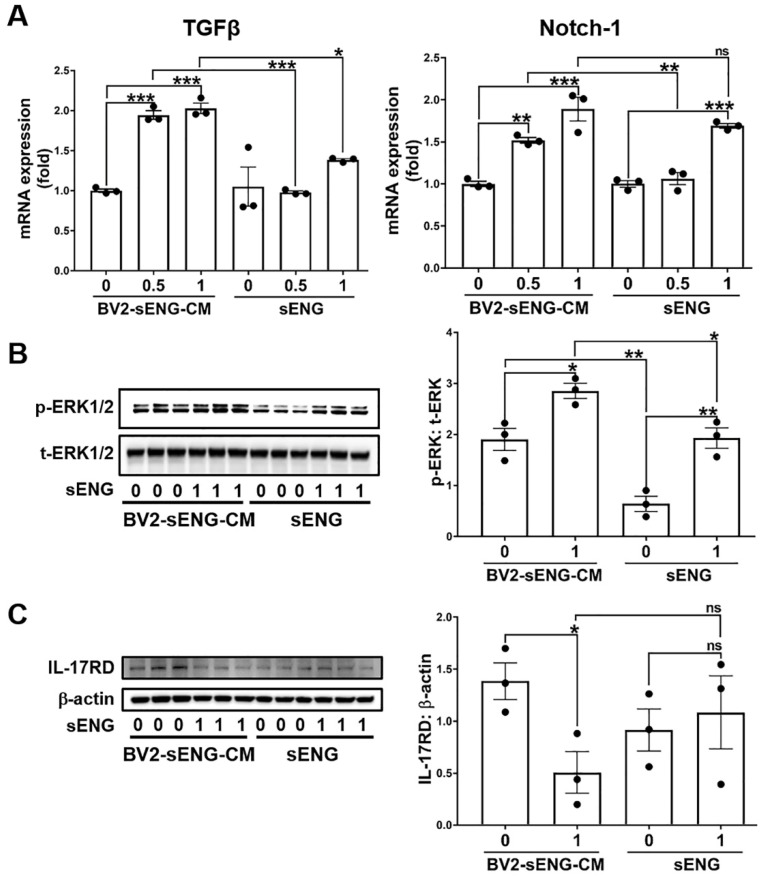
Soluble ENG-stimulated microglia regulate the expression of angiogenic mediators in endothelial cells. (**A**) Gene expression of angiogenic mediators in ECs treated with sENG (0.5 or 1 μg/mL)-treated microglia conditioned medium (BV2-sENG-CM). Mouse brain vascular endothelial cells (ECs) were incubated in a medium containing BV2-CM, BV2-sENG-CM, PBS, or sENG for 24 h. Data are presented as mean ± SEM; * *p* < 0.05, ** *p* < 0.01, *** *p* < 0.001; one-way ANOVA. (**B**) BV2-sENG-CM or sENG induced p-ERK1/2 expression in ECs. *n* = 3; data are presented as mean ± SEM; * *p* < 0.05, ** *p* < 0.01; one-way ANOVA. (**C**) BV2-sENG-CM decreased IL-17RD expression in ECs. *n* = 3; data are presented as mean ± SEM; * *p* < 0.05; one-way ANOVA.

**Figure 5 ijms-23-01225-f005:**
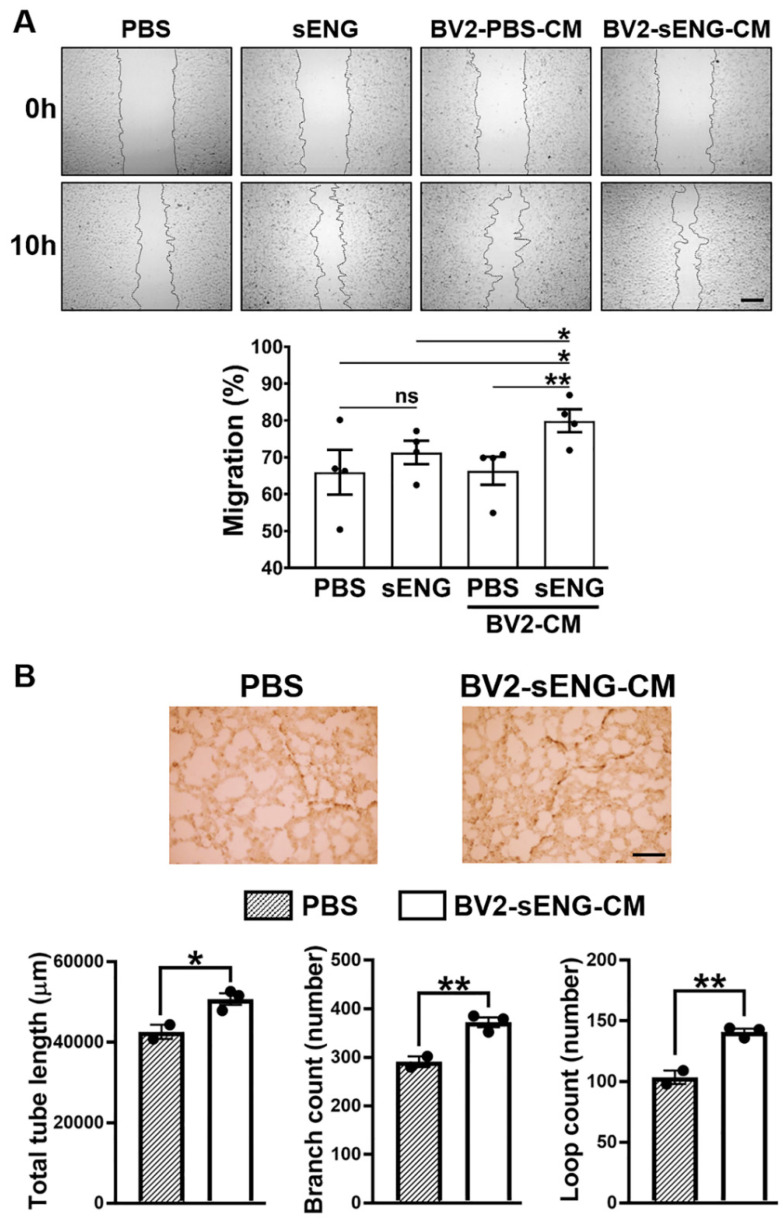
Soluble ENG-stimulated microglia induces a hyper-angiogenic phenotype of endothelial cells. (**A**) BV2-sENG-CM induced EC migration in a scratch-wound assay. The rate of EC migration in each treatment was quantified by measuring the distance between the edges of scratched area at 10 h compared to 0 h after the scratch. *n* = 3; data are presented as mean ± SEM; * *p* < 0.05, *** p* < 0.01; scale bar: 400 μm. (**B**) BV2-sENG-CM increased endothelial tube formation. The tube length, branch count, and loop count were automatically and blindly measured using Ibidi software. Three independent experiments were performed, and representative images are shown. *n* = 3; data are presented as mean ± SEM; * *p* < 0.05, *** p* < 0.01, Student’s *t*-test; scale bar: 500 μm.

**Figure 6 ijms-23-01225-f006:**
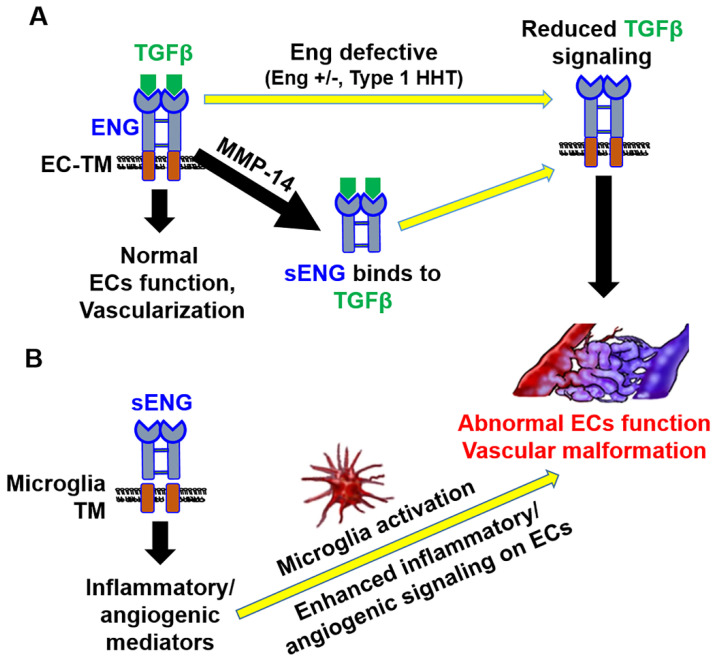
The role of soluble ENG in endothelial dysfunction and vascular malformation. **(A)** Endoglin (ENG) mediates the TGFβ signaling pathway in endothelial cells (ECs). The mutation of Eng (haploinsufficiency) reduces TGFβ signaling. MMP-14 cleaves to the extracellular domain of ENG in ECs, producing a soluble form of ENG (sENG). The sENG acts as a decoy receptor for the TGFβ, resulting in impaired TGFβ signaling. Therefore, both mutant ENG and sENG lead to the disruption of TGFβ signaling and subsequently mediate EC dysfunction and vascular malformation. TM, transmembrane. (**B**) Soluble ENG stimulates microglia (by an unknown mechanism) and leads to microglial activation with up-regulation of inflammatory/angiogenic mediators. The microglia-derived inflammatory/angiogenic mediators possibly induce hyper-angiogenic signaling on ECs and cause EC dysfunction. Taken together, the sENG-induced disruption of TGFβ signaling in ECs and the activation of microglia may work together to drive vascular malformations.

## Data Availability

All data generated for this study are available from the corresponding authors upon reasonable request.

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
