# Peer review of "Soluble Endoglin Stimulates Inflammatory and Angiogenic Responses in Microglia That Are Associated with Endothelial Dysfunction"

_ijms, 2022, doi:10.3390/ijms23031225_

Round 1
Reviewer 1 Report
Eun S. Park et al. in this work investigated the role of microglia in the sENG-induced endothelial dysfunction in bAVM pathology. The authors have shown that systemic sENG injection in parallel with focal overexpression of VEGF activates microglia around dysplastic vessels. In particular, the sENG treatment caused the increase of NLRP3 expression in mouse brain. In addition, in in vitro experiments with BV2 cells, soluble ENG induced gene expression of angiogenic and inflammatory mediators. Moreover, conditioned medium from these cells significantly increased endothelial cells migration and tube formation. Even though there are clear evidences that suggest the contribution of microglia in the pathology of sENG-associated vascular malformations, the conclusions regarding the main mechanism by which sENG exerts its effect it is quite elusive and more solid experiments are required.
More principally, I have the following major concerns with the manuscript:
Figure 1B: Representative images of blue latex casted-clarified brains in mice injected with sENG and AAV-VEGF-A. Control images of mice treated with VEGF overexpression alone without sENG injection or sENG treatment with control AAV1-LacZ injection are missing. Please add them to the figure.
Figure 1B and 1D: The field of view in these insets is quite small, it is difficult to have a clear idea of the overall enlargement and abnormal vasculature of the region. Please replace these images with a wider field of view.
Figure 3B: It is shown that the treatment of BV2 cells with sENG alone causes the increase of NLRP3 expression level. However, in Figure 2A, in mouse brain, no NLRP3 positive cells were detected upon sENG treatment. Could the authors please explain this difference?
Figure4C: IL-17RD protein expression. Looking at the original blot, it seems that there was a problem during the development of this membrane (ECL?). Therefore, the reduction of the anti-angiogenic mediator IL-17RD could be only apparent. Please add a new western blotting for this protein with a better development/exposure.
In the abstract, (line 14) it is specifically stated that the underlying mechanism of sENG-induced vascular malformations is not clear. In the discussion section (line 209) the authors claim that the release of specific factors by sENG-treated microglia regulates angiogenic property of ECs. However, in the manuscript there is only a panel of genes related with angiogenic and inflammatory mediators quantified by RealTime. This is not sufficient to support the explanation of the mechanism that they propose. Supplementary experiments have to be done in this regard.
Line 268-270: incomplete sentence. Rephrase it.
Reviewer 2 Report
Dear Authors,
You present here an interesting work regarding the involvement of soluble endoglin in the development of brain arteriovenous malformations. Considering the gravity of this type of disease, the subject that you focus your research is of very high importance. The results obtained suggest also that microglia may contribute to soluble endoglin-induced endothelial dysfunction via releasing inflammatory and angiogenic factors.
I suggest that you use some anti-inflammatory and antiangiogenic drugs in your tests, in order to check if these can reduce the effect of soluble endoglin and if so, to verify the meachanism of action.
The English corrections that are needed, are minor.
Reviewer 3 Report
The manuscript "Soluble endoglin stimulates inflammatory and angiogenic responses in microglia that are associated with endothelial dysfunction" by Eun S. Park et al. is devoted to the study the pathology of blood vessels, blood circulation. In particular, arteriovenous malformations of the brain are the main cause of intracerebral hemorrhage in children and young people, as well as other neurological symptoms, however, this disease is poorly understood, and its treatment options are limited. In this article, the authors investigate the possible mechanisms of the effect of soluble endoglin on the formation of dysplastic vessels, as well as the role of microglia in the inflammation and dysfunction of endothelial cells caused by soluble endoglin. The manuscript can presents a great interest to the readers of Molecular Sciences, however, at the moment it requires some minor modifications.
In the work, the authors selected suitable methods, thanks to which they managed to achieve the goals set for themselves. However, there are a few comments that should be taken into account:
- In Figures 1 CD and 2B-C, for a more visual perception of information, the authors use the display of individual values in the form of dots on the graphs. Perhaps it would be necessary to bring all the drawings into a single view and add dots to the rest of the graphs.
- In the captions to all figures displaying histograms, the authors should indicate in what form the bars representing the range of values +/- (Standard Error or Standard Deviation) are presented.
- It is also necessary to specify using which criteria the statistical pvalues were determined when using One-way ANOVA.
- The text of the manuscript lacks one of the key sections of any experimental/review article - the conclusion, thanks to which the reader could get information about the main theses of this study.
Also, in the future, I would like to advise the authors to use a wider sample of experimental animals to prove their hypotheses.
Round 2
Reviewer 1 Report
Accept in present form.